# A Facile Synthesis of (PIM-Polyimide)-(6FDA-Durene-Polyimide) Copolymer as Novel Polymer Membranes for CO_2_ Separation

**DOI:** 10.3390/membranes9090113

**Published:** 2019-08-31

**Authors:** Iqubal Hossain, Abu Zafar Al Munsur, Tae-Hyun Kim

**Affiliations:** 1Organic Material Synthesis Laboratory, Department of Chemistry, Incheon National University, Incheon 22012, Korea; 2Research Institute of Basic Sciences, Incheon National University, Incheon 22012, Korea

**Keywords:** CO_2_ separation, random copolymer, PIM-polyimide, permeability-selectivity, pressure effect

## Abstract

Random copolymers made of both (PIM-polyimide) and (6FDA-durene-PI) were prepared for the first time by a facile one-step polycondensation reaction. By combining the highly porous and contorted structure of PIM (polymers with intrinsic microporosity) and high thermomechanical properties of PI (polyimide), the membranes obtained from these random copolymers [(PIM-PI)-(6FDA-durene-PI)] showed high CO_2_ permeability (>1047 Barrer) with moderate CO_2_/N_2_ (> 16.5) and CO_2_/CH_4_ (> 18) selectivity, together with excellent thermal and mechanical properties. The membranes prepared from three different compositions of two comonomers (1:4, 1:6 and 1:10 of x:y), all showed similar morphological and physical properties, and gas separating performance, indicating ease of synthesis and practicability for production in large scale. The gas separation performance of these membranes at various pressure ranges (100–1500 torr) was also investigated.

## 1. Introduction

CO_2_ separation using polymeric membranes has attracted much interest due to their easy process-ability, low energy consumption, low capital and maintenance cost, module compactness and environmental-friendliness relative to other separation techniques, including cryogenic distillation, amine scrubbing, and pressure swing adsorption [1,2,3,4,5]. To this end, various types of polymer materials have been developedto remove CO_2_ from low-quality natural gas, syngas, and flue gas [2].

The key parameters used to address the intrinsic gas separation properties of specific membrane materials are their permeability (*P*) and selectivity (*α*). Briefly, permeability is a product of diffusivity (*D*) and solubility (*S*) coefficients (i.e., *P = D x S*).The ideal selectivity (*α_A/B_*) is the ratio of permeability between two single gases (*P_A_* and *P_B_*), which can be depicted as α_A/B_ = *P_A_/P_B_*. The desired membrane material for gas separation in industrial applications must have both high permeability and high selectivity. However, it is very difficult to realize both these properties in the same material. In fact, dense polymeric membranes inevitably suffer from an undesired trade-off between gas permeability (flux) and selectivity (efficiency)typically knownas the ‘Robeson upper-bound relationship’ which was recently verified theoretically [6,7,8]. The Robeson upper-bound relationship is widely accepted as an empirical guidelines by which to determine the overall performance of a newly developed polymeric membrane. With this, design of a polymer membrane with the ultimate aims of high permeability and selectivity has become the goal of researchers in this field.

The permeability of a polymer membrane is usually facilitated by the incorporation of flexible polymers like polydimethylsiloxane(PDMS) [9] or a rigid polymer with a conjugated backbone like poly[(1-trimethylsilyl)-1-propine] (PTMSP) [10], but their poor selectivity and mechanical properties limited their utility in gas separation applications. Alternatively, high permeability in polymer membranes isoften realized by employed glassy polymers with rigid structures, they display high degrees of diffusivity-selectivity (*D*_A_/*D*_B_, where *D_A_* and *D_B_* refer to the diffusivity coefficient to gas A and B) caused by narrow distributions of free volume. Excellent mechanical properties are another feature of these rigid polymers with porous structures.

Meanwhile, the selectivity relies heavily on the solubility-selectivity and diffusivity-selectivity, *α_A/B_* = *S*_A_/*S*_B_ × *D*_A_/*D*_B_ (*S_A_* and *S_B_* are the solubility coefficient to gas A and B). Therefore, enhanced gas selectivity can be obtained via a combination of enhanced gas diffusivity selectivity and/or increased solubility of the particular gas in the polymer. Here, the diffusivity selectivity is correlated with thesize-sieving capability of the polymer material, and relies on the mobile character of the target gas molecule compared to those of other gas molecules, the structural factors such as the rigidity of the polymer chain and the extent of inter-segmental polymer chain packing [4]. The solubility selectivity, on the other hand, relies on the interactions of the polymer with the penetrant molecule and also on the condensability degrees of the penetrants to the polymer.

Among several polymer materials commercialized for gas separation, polyimides (PI)s have been most widely used because of their excellent physicochemical properties (e.g., thermal and mechanical, and chemical stability [4,10,11,12,13,14,15,16] with a moderate level of permeability. Moreover, these properties, including the permeability levels, can easily be changed by varying their structures using proper monomers. For example, PI membranes obtained from 4,4’-hexafluoro-isopropylidene di-phthalic anhydride (6FDA) exhibiteda high CO_2_/CH_4_ separating capability because of suppression of the intra-segmental mobility [4,17]. Nevertheless, most of the polymers with a rigid backbone structure, including PIs, do not satisfy the industrial requirements of high CO_2_/lightgas selectivity, nor of high CO_2_ permeability.

Polymers of intrinsic microporosity (PIMs) have emerged as a novel polymeric membrane material for gas separation in the last decade [3,18,19,20,21]. Having a fused-ring and ladder-type contorted conformation, the unique structure of PIMs interrupt the packing of polymer chains in the solid-state, causing high free volume and a low surface area. Accompanying these properties is excellent gas separation performance that surpasses Robeson’s upper bounds [19,20]. 

Several efforts have been put forth to combine the characteristic high free volume of PIMs with the excellent physicochemical properties of PIs, and therefore, several PIM-polyimide (PIM-PI) homopolymers and copolymers have been attempted for gas separation [2,3,22,23,24,25]. Some of these PIM-PI copolymer membranes, in particular the multiblock-type copolymer of PIM-PI and a 6FDA-durene-based PI [(PIM-PI)x-*b*-(PI)y], proved to have excellent thermal and mechanical properties [2]. They also exhibit excellent gas separation performance, positioning them well above the Robeson upper bound [2]. However, their multistep synthesis and critical control of the chain length of each block in the block copolymer remains challenging and limits their practical application for industry. Moreover, the morphology of a block copolymer is heavily dependent on the size of the individual block lengths. Even small changes of block length can affect the morphology dramatically and cause changes in the properties of the corresponding membranes. Therefore, development is still needed of more easily preparable and scalable polymers that combine the superb thermomechanical stabilities of PIs with the rigid and contorted structures of PIMs. This should increase the molecular sieving property, thereby enhancing the permeability to and selectivity for various gas pairs.

We report herein, a straightforward one-step preparation of random-type copolymers between PIM-PI and 6FDA-durene-based PI, designated as [(PIM-PI)x-(6FDA-durene-PI)y], with three different PIM-PI compositions (1:4, 1:6 and 1:10). We investigated the properties of the corresponding polymer membranes for CO_2_ separation and performed copolymer synthesis on a 100 g scale to confirm the ease of membrane synthesis and productivity.

## 2. Materials and Methods

### 2.1. Synthesis of [(PIM-PI)x-(6FDA-durene-PI)y] Copolymers (1) with Different PIM-PI Compositions

Atwo-step synthesis (that is, polyamic acid preparation followed by imidization) was carried out to prepare the [(PIM-PI)x-(6FDA-durene-PI)y] random-type copolyimides (1) with three different compositions following the literature [4,15,26].

#### Synthesis of (PIM-PI)-(6FDA-Durene-PI)(1:4)

General procedure for the synthesis of the *(PIM-PI)-(6FDA-durene-PI)(1:4):* a mixture of dianhydridemonomer (2) (23.0 g, 36.59 mmol, 22), 6FDA (3) (65.00 g, 146.36 mmol, TCI, Seoul, Korea) and durene (4) (30.05 g, 182.95 mmol, TCI, Seoul, Korea) in 150 mL of DMAc were stirred in a dried 1000 mL round bottom flask equipped with a condenser under nitrogen gas for 3 h at ice bath temperature, followed by room temperature for 12 h to form the poly(amic acid). Acetic anhydride (41.0 g, Sigma Aldrich, Yongin, Korea) and trimethylamine (40.69 g, Sigma Aldrich, Yongin, Korea) were then added to this reaction mixture and the temperature of the reaction mixture was increased to 110 °C for 3 h to induce the complete imidization of poly(amic acid) into the corresponding polyimide. The highly viscous solution was then obtained by cooling to room temperature, and was then diluted with DMAc (50 mL) and precipitated into methanol (2.0 L, DaeJung Chemicals, Shiheung-city, Korea) with stirring. Brownish-yellow polymer beads were separated by filtration, followed by washing with methanol and deionized water several times, and finally dried in a vacuum oven at 80 °C for 48 h to obtained the desired (PIM-PI)-(6FDA-durene-PI)(1:4) copolyimide with a high yield (92%); *δ*_H_ (400 MHz, CDCl_3_); 8.09–8.02 (8H, br signal, ArH), 8.01–7.90 (16H, br signal, ArH), 7.41–7.37 (2H, br s, ArH), 7.32–7.27 (2H,br s, ArH), 6.75–6.67 (2H, brs, ArH), 6.41–6.33 (2H, brs, ArH), 2.43–2.25 (4H, m, CH_2_), 2.20–1.92 (60H, br signal, ArCH_3_), 1.38–1.23 (12H, m, CCH_3_); GPC (CHCl_3_, RI)/Da *M*_w_ 4.7 × 10^4^, PDI = 2.2; ATR-IR (cm^−1^) 3078, 2964, 2924 & 2864 (C-H stretching), 1784 (C=O asymmetric stretching of imide), 1720 (C=O symmetric stretching of imide), 1624, 1422 (C=C), 1352 (C-N stretching of imide), 1256 (C-F stretching), 1209, 1192, 1108 (C-O-C), 748(ring deformation of imide) and 227 (C-N out of plane bending).

Copolymer (PIM-PI)-(6FDA-durene-PI)(1:6) and (PIM-PI)-(6FDA-durene-PI)(1:10) were prepared usingthe same method as (PIM-PI)-(6FDA-durene-PI)(1:4) with different monomer ratios. In brief, 15.3 g and 9.2 g of monomer (2); 28.2 g and 26.4 g of durene (3) with 65.0 g of 6FDA (3) were used for the synthesis of (x:y)(1:6) and (x:y)(1:10) composition, respectively. IR characteristic absorption peaks of both compositions were similar as that of (PIM-PI)-(6FDA-durene-PI)(1:4) with only a slight difference in the intensity. For the (x:y)(1:6) composition: Yield (93%); δ_H_ (400 MHz, CDCl_3_) 8.09–8.02 (12H, br signal, *ArH*), 8.01–7.90 (24H, br signal, *ArH*), 7.41–7.37 (2H, br s, ArH), 7.32–7.27 (2H, br s, *ArH*), 6.75–6.67 (2H, br s, *ArH*), 6.41–6.33 (2H, br s, *ArH*), 2.43–2.25 (4H, m, *CH_2_*), 2.20–1.92 (84H, br signal, *ArCH_3_*), 1.38–1.23 (12H, m, *CCH_3_*); GPC (*CHCl_3_*, RI)/Da Mw 4.88 × 104, PDI = 1.74. For the (x:y)(1:6) composition: δ_H_ (400 MHz, *CDCl_3_*) 8.10–8.04 (20H, br signal, *ArH* ), 8.01–7.91 (40H, br signal, *ArH*), 7.39–7.33 (2H, br s, *ArH*), 7.32–7.27 (2H, br s, *ArH*), 6.75–6.67 (2H, br s, *ArH*), 6.41–6.33 (2H, br s, *ArH*), 2.33–2.25 (4H, m, *CH_2_*), 2.23–1.98 (132H, br signal, *ArCH3*), 1.44–1.25 (12H, m, *CCH_3_*); GPC (*CHCl3*, RI)/Da Mw 4.27 × 104, PDI = 1.26.

### 2.2. Membrane Preparation

Each polymer membrane was prepared using a solution (~3 wt% in CHCl_3_) casting technique into a flat-bottomed glass dish, followed by slow evaporation of the solvent under a minimum flow of N_2_ or Ar at ambient temperature as follows. The corresponding copolyimides with different compositions (1) were dissolved in CHCl_3_ (~3% *w*/*v*, g/mL), stirred at room temperature overnight and then filtered. Each polymer solution was carefully poured into a glass dish covered with aluminum foil having small holes and the solvent was evaporated in N_2_ or Ar gas at r.t for three days. It was then placed in an oven and the solvent completely dried at 70 °C for 24 h. The dry membranes were obtained, cooled to room temperature, peeled away from the glass-plate, dried again at 70 °C in an oven for 24 h, and finally stored at ambient temperature. The thickness of all the membranes was controlled to fall within the range 45 to 55 µm.

### 2.3. Characterization Methods and Measurement of Gas Separation Performance of the Membranes

For characterizations and gas separation methods, we followed the previous report [2,4], which is detailed in the Appendix A.

## 3. Results

### 3.1. Synthesis and Characterization of the [(PIM-PI)x-(6FDA-Durene-PI)y] Copolymers (1)

The random-type copolymers composed of both PIM-PI and 6FDA-durene-based PI (1) were synthesized and were designated as [(PIM-PI)x-(6FDA-durene-PI)y]. Three different PIM-PIcompositions (1:4, 1:6 and 1:10) of PIM-PI and 6FDA-durene-based PI were prepared by the polycondensation reaction between two dianhydride (2 and 3) and a diamine (4) monomer by targeting the monomer ratios as (1:4), (1:6) and (1:10). These were termed (PIM-PI)-(6FDA-durene-PI)(1:4), (PIM-PI)-(6FDA-durene-PI)(1:6) and (PIM-PI)-(6FDA-durene-PI)(1:10), respectively (Scheme 1).

The structure and the composition of each copolymer, (PIM-PI)-(6FDA-durene-PI)(1:4) and (PIM-PI)-(6FDA-durene-PI)(1:6), were determined by comparing the integral ratio between two comonomers: for example, the peak integration corresponding to the phenyl proton of 6-FDA (H_a,b,c_) was compared with the peak integration of the spirobisindane-based phenyl protons (H_1,2,3,4_), and were found to be 1:4, 1:6 and 1:10 (Figure 1). In fact, the actual compositions between two comonomers were assigned to be the same as our original feed ratio calculated by theoretical calculation, indicating the reliability and validity of our random copolymer preparation. Consequently, large scale synthesis of these [(PIM-PI)_x_-(6FDA-durene-PI)_y_] copolymers (1) on 100 g scale was attempted and was successful.

The structures of [(PIM-PI)_x_-(6FDA-durene-PI)_y_] copolymers (1) were further confirmed by ATR-IR spectroscopic analysis (Figure 2), by showing peaks around 1780 and 1720 cm^−1^ corresponding to both asymmetric and symmetric stretching for imide C=O absorption, and peaks at 1350 (imide C–N stretching), 1256 (C–F), 742 (imide ring deformation) and 726 (C–N out of plane bend), all indicating successful synthesis of the desired copolyimides.

The molecular weights (M_w_) of the three [(PIM-PI)_x_-(6FDA-durene-PI)_y_] copolymers were determined by gel permeation chromatography (GPC), and were found to be in the range 43−49 kDa, supporting the achievement of copolymerization (Table 1).

### 3.2. Preparation of the [(PIM-PI)x-(6FDA-Durene-PI)y] Copolymer Membranes 

The [(PIM-PI)_x_-(6FDA-durene-PI)_y_] copolymer membranes with three different compositions, denoted as (PIM-PI)-(6FDA-durene-PI)(1:4), (PIM-PI)-(6FDA-durene-PI)(1:6) and (PIM-PI)-(6FDA-durene-PI)(1:10), were prepared using the solution-casting method with CHCl_3_ solution of each polymer to give flexible membranes with thickness of ~45–55 μm (Figure 3). All three membranes showed good solubility in organic solvents such as CH_2_Cl_2_, CHCl_3_, THF, and DMF, but were sparingly soluble in DMSO.

### 3.3. Morphological Analysis by XRD and AFM 

The morphologies of the three copolymer membranes were investigated using wide-angle X-ray diffraction (WAXD) and atomic force microscopy (AFM) in order to determine crystallinity and intermolecular chain distance of each polymer.

Figure 4 shows the WAXD data of the [(PIM-PI)-(6FDA-durene-PI)(1:4), (PIM-PI)-(6FDA-durene-PI)(1:6) and (PIM-PI)-(6FDA-durene-PI)(1:10)] copolymer membranes. Similar patterns of broad peaks were observed for all membranes, suggesting that their structures were amorphous. The polymer-polymer chain distance (or d-spacing), was further estimated using Bragg’s law, d = λ/2 sinθ (λ: the wavelength of 1.789 Å; θ: the scattering angle) (Table 1). Again, similar d-spacing values [7.03 Å for (PIM-PI)-(6FDA-durene-PI)(1:4), 6.98 Å for (PIM-PI)-(6FDA-durene-PI)(1:6) and 6.92 Å for (PIM-PI)-(6FDA-durene-PI)(1:10)] were observed. These results suggest that the gas separation properties of both membranes would be similar with only small variation. In addition, an additional peak at around 12.7 Å was observed for (PIM-PI)-(6FDA-durene-PI)(1:4) and (PIM-PI)-(6FDA-durene-PI)(1:6), which had more PIM-PI unit content. This additional peak is thought to arise from the distance between neighbouring spiro-carbon atoms in the PIM structure [2,27], but for the (PIM-PI)-(6FDA-durene-PI)(1:10), the peak was not obtained due to its very low PIM-PI content.

Moreover, similar morphologies without phase separation were also observed for all three copolymer membranes using AFM (Figure 5a–c), indicating that the highly permeable microporous PIM-PI unit is randomly distributed over the less permeable 6FDA-durene-PI domain. 

From the above morphological analyses, it was expected that the gas separation performance of the copolymer compositions (PIM-PI)-(6FDA-durene-PI)(1:4) and (PIM-PI)-(6FDA-durene-PI)(1:6) would be very similar, with slightly higher permeability for the former. In contrast, the gas permeability of (PIM-PI)-(6FDA-durene-PI)(1:10) composition was expected to be lower than the other two compositions based on the comparatively lower d-spacing in its microstructure. Moreover, the separation performances of all copolymer membranes is expected to follow the simple mixing rule [28,29,30] [lnP = φ1 lnP1 + φ2 lnP2, where φ is the volume fraction of each polymer] forthe random-type copolymers.

### 3.4. Gas separation Performance of the Copolymer Membranes 

The pure gas separation properties of the obtained three [(PIM-PI)x-(6FDA-durene-PI)y] (x:y = 1:4, 1:6 and 1:10) copolymer membranes were determined at 30 °C and 2 atm pressure using the constant-volume/variable-pressure method (Table 2) by monitoring the increase of downstream pressure upto 2 Torr (2.7 × 10^−3^ bar).

It is apparent that the copolymers with all three different PIM-PI compositions [(PIM-PI)-(6FDA-durene-PI)(1:4), (PIM-PI)-(6FDA-durene-PI)(1:6) and(PIM-PI)-(6FDA-durene-PI)(1:10)] exhibited higher CO_2_ permeability (> 1047 Barrer) relative to other gases (i.e., N_2_ and CH_4_) as follows: CO_2_>> N_2_> CH_4._ This order in permeabilities can be explained by the diffusivity, which takes the same order: CO_2_>> N_2_> CH_4_, as shown in Table 3 and Figure 6. Thus, the order of diffusivity is determined by the kinetic diameter of the penetrant gases, which is inversely proportional to the gas molecule size [CO_2_ (3.30 Å) < N_2_ (3.64 Å) < CH_4_ (3.80 Å)]; that is, smaller gases diffuse through the polymer matrix faster than larger gases do [31,32].

It is noteworthy that the performance of our newly-developed (PIM-PI)x-(PI)y copolymer membranes is much higher than most of the PIs reported in the literature [4,5,10,17], due to the incorporation of the highly permeable microporous PIM unit, resulting in enhanced solubilities of the penetrants in the newly-developed copolymers. For example, the solubilities of all the gases in the present studies were much higher than those of 6FDA-durene-PI (S = 0.17, 0.02 and 0.06 for CO_2_, N_2_ and CH_4_, respectively), which is a highly permeable PI [4].

When we compare the gas separation properties of the (PIM-PI)x-(6FDA-durene-PI)y copolymer membranes according to the three different compositions (1:4, 1:6 and 1:10), we find that two polymers (1:4 and 1:6) show very similar permeability and selectivity. Slightly higher CO_2_-permeability and selectivity were observed for (PIM-PI)-(6FDA-durene-PI)(1:4), which has higher PIM-PI content, than for (PIM-PI)-(6FDA-durene-PI)(1:6) (Table 2). It was expected that the highly microporous PIM unit would not only enhance the permeability, but would also increase the selectivity due to its enhanced solubility as well as the diffusivity of this unique structure. Nevertheless, the difference in the permeability between the two copolymers (1:4 and 1:6) was small, and this is ascribed to the random distribution of the highly permeable unit (PIM-PI) of the copolymer in the low permeable domain. This assumption is supported by morphological analyses using both XRD and AFM. On the other hand, the permeability of the (PIM-PI)-(6FDA-durene-PI)(1:10) polymer showed lower permeability than the other two compositions did due to the lowest content of highly permeable PIM-PI in its structure. Nevertheless, the permeability of (PIM-PI)-(6FDA-durene-PI)(1:10) was still high (1047 Barrer) compared to other rigid PI-type polymers reported. This is certainly a great advantage of random-type copolymers because, as mentioned earlier, the changes of the morphology and the physical properties, together with gas separation performance, according to changes in the composition are significant for the typical block-type copolymers [2].

Moreover, the present random-type PIM-PI copolymers displayed moderate selectivity of CO_2_/CH_4_, which is a common feature of most glassy polymers (e.g., PIs, PES, PEK and PPO), in which diffusivity selectivity has more influence than solubility selectivity does (Table 3). On the other hand, the moderate permselectivity of CO_2_/N_2_ was mainly influenced by solubility selectivity (Table 3), which feature is similar in most of the PIM-based glassy polymer membranes. Therefore, the perceptive combination of these two different compositions mutually enhanced the selectivity of the two pairs of gases: CO_2_/CH_4_ and CO_2_/N_2_.

It can be seen from Table 2 and Figure 7 that the permeability coefficient for all the gases decreased with increasing the amount of 6FDA-durene content. The same trend was observed for the selectivity of CO_2_/N_2._ These results are almost similar to the values predicted by the mixing rule [28,29,30]: lnP = φ_1_ lnP_1_ + φ_2_ lnP_2_, where φ is the volume fraction, P is the permeability coefficient and subscripts 1 and 2 indicate the corresponding homopolymers.

Nevertheless, the selectivity of CO_2_/CH_4_ does not maintain the trade-off relationship. Instead, the selectivity increased initially with the incorporation of 6FDA-durene, which was then decreased accordingly:(x:y)(1:4)> (x:y)(1:6)> (x:y)(1:10). This unusual trend may be due to the enhancement of CO_2_/CH_4_ diffusivity selectivity by the sudden decrease of CH_4_ diffusivity (which affects more for the gases with a higher kinetic diameter than the gases with a lower kinetic diameter) by the incorporation of 6FDA-durene. The trend was then followed by the gradual decrease of perm-selectivity, possibly due to the much decreased CO_2_ solubility.

### 3.5. Permeability vs Selectivity

As mentioned earlier, the trade-off relationship between permeability (P) and ideal selectivity (α) is a common phenomenon for most polymer membrane-based gas separation (i.e., higher permeability is obtained at the cost of reduced selectivity and vice versa). The CO_2_ permeability versus the CO_2_/CH_4_ (Figure 8a) and CO_2_/N_2_ (Figure 8b) selectivity values of the newly developed [(PIM-PI)x-(6FDA-durene-PI)y] membranes were then placed at the upper bound of the Robeson plots [6,7] and the modified upper bound of 2019 [33], and compared with the reported values obtained from the typical PIs [4,5,10,15,17], together with PIM-PI type homo- and copolymers [3,23,24,25]. The comparison results showed that our random-type [(PIM-PI)x-(6FDA-durene-PI)y] copolymers outperformed most of the PIs and PIM-PI homopolymers (Figure 8a). Furthermore, the trade-off results of the [(PIM-PI)x-(6FDA-durene-PI)y] copolymers crossed the upper bound line of 1991 for CO_2_/CH_4_, although they were still below the upper bound limit of 2008, and the modified upper bound of 2019.

In contrast, the separation performance of all random-type copolymer membranes for CO_2_/N_2_ fell below the upper bound 2008, and the modified upper bound of 2019 which is mainly redefined based on the outstanding performance of triptycene-based PIM materials, but still showed better performance than PIs did (Figure 8b).

### 3.6. Pressure Effect on Permeability

Due to its having the best gas separation performance, the (PIM-PI)-(6FDA-durene-PI)(1:4) copolymer membrane was further investigated for the effect of feed pressure on its gas-separation ability over a gas feed pressure range of 100–1500 Torr (0.1–2 bar) at 30 °C (Figure 9). We observed that the CO_2_ permeability increased dramatically with decrease in the pressure, whereas the CH_4_ and N_2_ permeability slightly increased with decrease in the feed pressure (Figure 9a). Therefore, the decrease in the CO_2_/CH_4_ and CO_2_/N_2_ selectivity that occurred at higher pressure was mainly due to decreased CO_2_ permeability (Figure 9b). A similar result was reported elsewhere [27,34,35,36,37]. This trend of gradual decrease in permeability with increasing pressure was also observed for other microporous rigid polymers and is ascribed to the filling of the sorption sites in the Langmuir model at a higher pressure range [35,36,37,38]. The separation performance at very low pressure is a good indication of the possible applicability of our materials for CO_2_ separation application.

All the gas separation properties strongly indicate that the newly developed [(PIM-PI)x-(6FDA-durene-PI)y] random-type copolymers are promising candidate materials to be used for CO_2_ separation, in particular at, but not only limited to, low pressures. 

### 3.7. Thermal and Mechanical Properties

Finally, the thermal and mechanical properties of the (PIM-PI)x-(6FDA-durene-PI)y copolymer membranes were investigated (Table 4 and Appendix A). 

TGA analysis showed that both copolymers had high thermal stability with initial decomposition temperatures above 465 °C, with the maximum weight loss temperature and residual weight loss, T_max_ and RW, at 536 °C and > 53%, respectively. This indicates excellent thermal stability, suitable for applications requiring high-temperature CO_2_ separation, of our (PIM-PI)x-(6FDA-durene-PI)y copolymer membranes (Appendix A and Table 4). 

The mechanical stability of the copolymer membranes was further analysed by measuring the stress-strain values of the all membranes at 50% RH (Appendix A and Table 4). All three membranes exhibited excellent tensile strength of more than 70 MPa higher than PIM-PI-1 homopolymer [24]. Incorporation of the 6FDA-durene-based PI unit onto the PIM-PI unit was, therefore, believed to enhance the mechanical stability of the corresponding [(PIM-PI)x-(6FDA-durene-PI)y] copolymer membranes, as we originally intended.

## 5. Conclusions

A series of random-type [(PIM-PI)x-(6FDA-durene-PI)y] copolymers with different composition (x:y = 1:4, 1:6 and 1:10) were synthesized for the first time using a facile one-step polycondensation reaction. By combining the highly porous and contorted structure of PIM with the excellent thermomechanical properties of polyimide, the membranes obtained from these random-type [(PIM-PI)*x*-(6FDA-durene-PI)y] copolymers showed very high CO_2_ permeability (> 1047 Barrer) with moderate CO_2_/N_2_ (over 16.3) and CO_2_/CH_4_ (over 18) selectivity, together with excellent thermal and mechanical properties. The membranes prepared from three different compositions of comonomers (1:4, 1:6 and 1:10 of x:y), showed similar morphological and physical properties, and only slightly different gas separation performance. This indicates ease of synthesis and practicality for large-scale production. Last but not least, the fruitful result of separation performance over a wide pressure range (100–1500 torr) also makes the new polymer appear to be a competitive material for applications requiring CO_2_ separation.

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
