# Peer review of "A Facile Synthesis of (PIM-Polyimide)-(6FDA-Durene-Polyimide) Copolymer as Novel Polymer Membranes for CO2 Separation"

_membranes, 2019, doi:10.3390/membranes9090113_

Round 1

Reviewer 1 Report

The authors describe the synthesis, characterization and transport properties of a set of three random copolymers with different ratios of a PIM-PI monomer and a traditional 6FDA-durene PI monomer. The study is carried out thoroughly and a number of supporting analysis (such as tensile tests and TGA) are carried out to further support the transport properties that form the core of the work. Unfortunately, the authors did not synthesize the two homopolymers, which would be essential for a fair conclusion on the effect of the copolymer composition. In spite of the thorough work, therefore, the manuscript gives nevertheless a strong impression of being incomplete.

In their discussions, the authors should include the properties of the two homopolymers, either from the literature (although that is sensitive to random error) or from novel synthesis.

Further comments:

Line 45-46: the authors define PTMSP as flexible. This is not true. PTMSP owes its high permeability to its RIGID conjugated backbone which inhbits efficient packing like in PIMs. Section 2.1.2 and 2.1.3: the procedure may be reduced to a minimum in order to avoid repetition, but the amounts of reagents used should be mentioned at least. Line 146: various literature reports confirm that PIMs tend to retain residual solvent, and therefore they are often treated in methanol. Is the drying at 70°C sufficient to remove the casting solvent? The TGA curves show a slight decrease before the onset of degradation. Can this be related to residual solvent, and how would that have affected the results? Tables 2,3 and 4 can be combined in a single table. Line 216-217, the expression “the highly porous permeable PIM-PI unit ….” is incorrect. The unit is not porous, but the unit causes inefficient packing of the polymer bulk, which therefore becomes microporous. Lines 218-222: what is the basis for these “expectations”? In an extensive discussion, Robeson discusses that for a homogeneous blend, the permeability should lie between the values of the pure polymers, according to the following relation: ln P = φ1 ln P1 + φ2 ln P2, where φ is the volume fraction of each polymer. The same equation should be valid for a random copolymer instead of a homogeneous blend. [Robeson, L.M. Polymer blends in membrane transport processes. Ind. Eng. Chem. Res. 2010, 49, 11859-11865] Please plot the data as a function of fraction of the two polymers and plot the data of the pure polymers (from the literature or from own synthesis) as well. As a first approximation, the weight fraction can be used, or the volume fraction can be calculated by the additive rule if the two individual densities are known) Table 2 and figure 6. Commonly, the trade-off between permeability and selectivity occurs but in this case the opposite is found. Please show the data of the pure polymers and explain if the interpolation between the values of the pure polymers is normal (see also previous point 6). Figure 5. The x-axis is inverted and plots the kinetic diameter from high to low instead of from low to high. This is confusing. Most papers in the literature use a logarithmic scale for the diffusion coefficient and plot the gas diameter from low to high, resulting in a decreasing trendline. Please follow this convention. Which gas diameter was used? Please provide the reference. Various other papers demonstrate that the diameters defined by Teplyakov and Meares [V. Teplyakov and P. Meares, Gas Sep. Purif., 1990, 4, 66-74] proves to give better correlations than the kinetic diameter. For instance PIs reported by Budd’s group [Macromolecules 2014, 47, 5595−5606, dx.doi.org/10.1021/ma5011183]. Figure 6: the value of PIM-PI homopolymer should be plotted in the same graph and discussed too. The 2008 upper bound is outdated. More recent upper bounds have been published for mixtures by Pinnau’s group [Wang et al., Mater. Today Nano. 3 (2018) 69–95. doi:10.1016/J.MTNANO.2018.11.003] and for pure gases by McKeown’s group [Comesaña-Gándara et al., Energy Environ. Sci. (2019). doi:10.1039/C9EE01384A]. Line 308-310: The conclusion “This excellent separation performance at low pressure probably originates from stronger interaction between CO2 and Langmuir cavities than with other gases due to its quadrupole moment and molecular size” is not correct. This is a simple effect of the dual mode sorption and of the decreasing solubility with increasing pressure. As demonstrated by the group of Budd, high affinity for CO2 leads to an inversion of the diffusion coefficients of CO2 and N2 because of ‘immoblization’ of CO2 by the strong interaction [Mason et al. Macromolecules 2014, 47, 1021−1029; Satilmis et al. Journal of Membrane Science 2018, 555, 483-496] and this is not the case in the present work (Table 3)

Minor corrections:

Line 28: module instead of modulus

Line 47: PDMS is the abbreviation of polydimethylsiloxane and not of polydimethylphenylsiloxane.

Table 2: the ratio 1:6 is reported twice

Table 3: the table reports diffusivity, but the column headers report PCO2, PN2 and PCH4

Line 245: please correct 510 to 5,10

Reviewer 2 Report

This paper presents some interesting results and it is overall well-written. 

A few typos - in Fig. 4 and Table 2 , the ratio (1:6) is written twice - I believe it should be (1:6) for one and (1:10) for the other case.  The authors only showed performance upto 1800 torr (30 psia) which is still very low (by almost a factor of 10) compared to typical pressure requirements in gas separation processes. One of the biggest drawbacks of polymer-based membranes is their tendency to plasticize at higher pressures. Having high performance values at such low pressures, unfortunately, is inconsequential to the actual applications. The authors should address this issue in the revised paper.  While the authors have compared their performance with PI membranes, it is not clear how these membranes perform against recently developed high-performance polymer membranes like fluoropolymer membranes, thermally rearranged polymers etc. The authors should try to provide such comparisons. 

Reviewer 3 Report

The study here presented is an interesting one and overall can be deemed worthy of publication after some revision.

The introduction is quite sufficient for the introduction of the topic and no major change should be applied.

Within the materials and method section a certain amount of confusion can be found. For the synthesis process, instead of three different paragraphs listing the characteristic peaks for each copolymer, a single paragraph, illustrating thoroughly the chemical protocol should be left. Within it, I would suggest to simply explicit the different quantities of reagents adopted for each synthesis.

The listing of the various peaks can be left for the results discussion part.

In the paragraph regarding the membrane preparation, the protocol is well described.

During the discussion of the characterization of the material, it would make more sense to have both the NMR and FTIR spectra (importing it from the supporting document), along with the corresponding peak analysis (removing the list from the materials and methods part).

Within the various tables and charts comparing the properties (e.g. permeability) it could be useful to also have a reference for the pure homopolymers (or better a range of common values), in order to better appreciate the influence of the copolymerization process.

In table 2, the last name of the copolymer should probably read (1:10) not (1:6), repeated from the line above. Moreover, in the description of the table and the permeation data charts, the downstream pressure should be specified (since it was changed in some tests).

English language is overall acceptable, even though a check should be conducted.

Round 2

Reviewer 1 Report

The authors have responded extensively to the comments of the reviewer. All critical issues were solved and the manuscript was significantly improved. The manuscript is now suitable for publication in membranes.

For the benefit of readability, I would only change the order of the data in Tables 1, 2 and 4: the two homopolymers should be plotted in first and last line of the table, in the order of increasing copolymer composition. This point can be solved instantaneously and needs no further revision of the manuscript.

Furthermore, there seems to be an issue with the notes in Table 2, which now appear next to the table and not under the table. This can be taken care of by the typesetter and does not need a further revision

Reviewer 2 Report

I believe the authors have addressed the primary concerns in their response to reviewers section. It would be better if the authors could incorporate some of these responses, especially their advantages over polymer counterparts since I think readers will have the same questions when they go through this paper. Overall, the paper can be accepted in its present form.